# Evaluating the Impact of Demographic Factors on Subject-Independent EEG-Based Emotion Recognition Approaches

**DOI:** 10.3390/diagnostics16010144

**Published:** 2026-01-01

**Authors:** Nathan Douglas, Maximilien Oosterhuis, Camilo E. Valderrama

**Affiliations:** 1Department of Applied Computer Science, University of Winnipeg, Winnipeg, MB R3B 2E9, Canada; douglas-n@webmail.uwinnipeg.ca (N.D.); oosterhuis-m@webmail.uwinnipeg.ca (M.O.); 2Department of Community Health Sciences, Cumming School of Medicine, University of Calgary, Calgary, AB R3B 2E9, Canada

**Keywords:** emotion recognition, electroencephalography, EEG signal processing, deep learning, attention-based fusion

## Abstract

***Background***: Emotion recognition using electroencephalography (EEG) offers a non-invasive means of measuring brain responses to affective stimuli. However, since EEG signals can vary significantly between subjects, developing a deep learning model capable of accurately predicting emotions is challenging. ***Methods***: To address that challenge, this study proposes a deep learning approach that fuses EEG features with demographic information, specifically age, sex, and nationality, using an attention-based mechanism that learns to weigh each modality during classification. The method was evaluated using three benchmark datasets: SEED, SEED-FRA, and SEED-GER, which include EEG recordings of 31 subjects of different demographic backgrounds. ***Results***: We compared a baseline model trained solely on the EEG-derived features against an extended model that fused the subjects’ EEG and demographic information. Including demographic information improved the performance, achieving 80.2%, 80.5%, and 88.8% for negative, neutral, and positive classes. The attention weights also revealed different contributions of EEG and demographic inputs, suggesting that the model learns to adapt based on subjects’ demographic information. ***Conclusions***: These findings support integrating demographic data to enhance the performance and fairness of subject-independent EEG-based emotion recognition models.

## 1. Introduction

This article extends our earlier conference paper [1] by incorporating additional analyses, including a demographic ablation study and an attention-weight interpretation, providing deeper insights into how age, sex, and nationality influence subject-independent EEG-based emotion recognition.

Emotions are regulated by the hippocampus, amygdala, and prefrontal cortex (PFC) [2]. These brain regions interact to associate emotional stimuli with memories and generate appropriate responses [3,4]. The interaction between the hippocampus, amygdala, and PFC can be measured indirectly using electroencephalography (EEG) [5]. This recorded electrical activity can be used to develop machine learning models to predict emotions [6,7,8,9]. Recent advances in artificial intelligence and multimodal learning have significantly improved EEG-based emotion recognition, with deep spatiotemporal fusion models demonstrating state-of-the-art results [10,11,12,13,14,15,16,17].

To develop EEG-based emotion recognition models, two approaches can be used: subject-dependent and subject-independent [18]. The subject-dependent approach involves using EEG signals from the same subjects for both the training and testing phases, necessitating that a subject’s data appears in both sets. In contrast, the subject-independent approach builds a model from a group of subjects’ EEG signals and tests it on a different individual not included in the training phase. From these two approaches, the subject-independent model is more practical, as it allows the development of a model that does not require recalibration to be used by a new user. However, given the high variability of EEG signals across individuals, models developed under the subject-independent approach tend to yield lower classification performance than those trained using the subject-dependent approach [19].

The problem occurring for the subject-independent approaches is known as the shift-domain problem [20], in which the assumption that the training and test set features are from the same distribution is violated. In the case of EEG, as signals from subjects exhibit different distributions, the features extracted from those features also display a different distribution. As a result, the patterns learned by the model from the training set may not fully apply to new individuals, thereby reducing the predictive performance of the models [21,22].

To address the domain-shift problem, one approach is to include the demographics of the subjects alongside EEG signals to train the model [5]. For instance, Li et al. showed that including the age and biological sex of the subjects in the last layer can improve emotion prediction [23]. Similarly, Peng et al. compared a model trained and tested using same-sex subjects with a model trained on subjects of the same sex and tested on subjects of the opposite sex (i.e., cross-sex model), reporting that same-sex models outperformed the cross-sex model for predicting emotions [24]. This strategy was also used in [25], where models trained with subjects from the same nationality were compared with cross-nationality models. Again, the models trained and tested on individuals from the same nationality yielded a higher predictive performance, thus suggesting that nationality influences the patterns extracted by the model for making the predictions. Finally, in [26,27], it was shown that including the subjects’ biological information and nationality can increase the likelihood of correctly predicting emotions.

Previous studies have demonstrated the potential of demographic information for improving the performance of subject-independent emotion recognition approaches. Incorporating factors such as sex, age, or nationality can enable the extraction of group-specific patterns that enhance prediction accuracy [28,29,30,31]. However, existing works present two main limitations. First, Li et al. [23] only considered age and sex within a dataset of subjects from the same nationality, thereby neglecting the influence of cultural background on emotion perception, a factor that previous research has identified as critical when comparing interdependent and individualistic societies [32,33]. Second, the comparisons made in [24,25] between same-sex and same-nationality models and their cross-sex and cross-nationality counterparts limit practical applicability, as demographic group membership may not always be known in advance. Moreover, this strategy requires training and maintaining multiple deep learning models tailored to each demographic subgroup, which is impractical in scenarios with limited data availability for specific populations. Therefore, further research is needed to develop a unified approach that integrates demographic information with EEG signals, enabling models to generalize across diverse populations irrespective of sex, age, or cultural background. Table 1 summarizes prior studies integrating demographic factors into EEG-based emotion recognition and highlights the methodological limitations that motivate our proposed approach.

To address these limitations, the present study proposes a deep learning framework that fuses EEG signals with demographic information to improve emotion recognition under a subject-independent setting. Specifically, we evaluate our approach using three benchmark emotion recognition datasets (SEED, SEED-FRA, and SEED-GER), which include EEG data from 31 participants of different sexes, ages, and nationalities. The main contributions of this work are as follows:We propose a unified deep learning framework that integrates EEG features with three key demographic variables (sex, age, and nationality) for emotion recognition.We analyze the relative influence of each demographic variable on recognition performance, providing insights into their individual and combined effects.We show that fusing demographic information enhances the generalization capability of subject-independent emotion recognition models across diverse populations.

## 2. Materials and Methods

### 2.1. Dataset

The study used three EEG datasets: SEED (SJTU Emotion EEG Dataset) [34], SEED-FRA [25] and SEED-GER [25] datasets, which are benchmark datasets commonly used for emotion recognition [5]. Table 2 describes these datasets. All datasets include signals recorded from 62 EEG channels using the 10–20 system EEG cap.

SEED contains EEG signals from 15 subjects who watched a series of film clips in Chinese. These clips were labeled as positive, neutral, and negative to capture emotional states. All subjects participated in three sessions, watching 15 clips each (5 per emotion). As a result, a total of around 45 EEG signals were collected for each subject.

SEED-FRA and SEED-GER are extensions of SEED, each consisting of 8 subjects who were shown clips in French and German, respectively. In SEED-FRA, each subject watched 21 videos per session across three sessions, resulting in a total of 63 EEG recordings per subject. In SEED-GER, each subject viewed 18 videos, yielding 54 EEG recordings per subject.

All datasets were class-balanced, having the same number of videos per emotion. Specifically, the number of videos for each emotion type was 15, 21, and 18 for SEED, SEED-FRA, and SEED-GER, respectively.

### 2.2. EEG Preprocessing

As EEG signals are prone to noise, we filtered them using a Butterworth filter within the range of 1–75 Hz. The selection of this range was made to ensure the inclusion of brain frequency bands (delta (δ): 1–4 Hz, theta (θ): 4–8 Hz, alpha (α): 8–12 Hz, beta (β): 12–30 Hz, and gamma (γ): 30–50 Hz) during the feature extraction process. Thus, this filter removed higher frequency components associated with eye blinking and motor functions.

### 2.3. Differential Entropy

The EEG signals were segmented using a four-second window. To calculate the DE for each window, it was assumed that the EEG signals followed a Gaussian distribution [35]. So, the DE was calculated as follows:(1)$$DE_{ch,b}=\frac{1}{2}\ln\left(\pi \times e\times \sigma_{ch,b}^{2}\right),$$
where ch was the $ch_{th}$ EEG channel (e.g., $FP_{1}$, $O_{2}$, $T_{1}$), *b* was the $b_{th}$ frequency band (δ, θ, α, β, or γ), and $\sigma_{ch,b}^{2}$ was the power spectrum density (PSD) for the $ch_{th}$ channel and the $b_{th}$ band. The PSD was computed through the discrete Fourier transform (DFT) as:(2)$$\sigma_{ch,b}^{2}=\frac{1}{N_{b}}\sum_{k=0}^{N_{b}}{\left(|X\left[k\right]|\right)}^{2},$$
where *X* was the DFT of the 4-s EEG window, and $N_{b}$ was the number of frequency components in the $b_{th}$ band.

Since DE was computed for each EEG channel and frequency band, this procedure resulted in a feature matrix of dimension 62 × 5 for each of the 4-s segments.

### 2.4. Standardization

Before model training, both EEG signals and demographic data were standardized. EEG signals were individually standardized for each session per subject using a z-score transformation:(3)$$z=\frac{\left(x-\mu \right)}{\sigma },$$
where *x* represents the DE feature vector, μ was the mean, and σ was the standard deviation computed per session per subject.

For the demographic data, one-hot encoding was applied to categorical variables, nationality, and gender. All demographic features were then standardized using the same z-score standardization technique to ensure consistency across variables.

### 2.5. Deep Learning Approaches

Two deep learning approaches were used to train the model. Approach 1 served as a baseline, consisting solely of the DE feature data. This model was used to establish a benchmark for the second approach. Approach 2 built on the first by incorporating the demographic information: sex (male or female), age, and nationality (Chinese, French, or German). The baseline model was used to evaluate the impact of adding such demographic data in the training process and overall performance.

All approaches were conducted on a workstation equipped with an AMD Ryzen Threadripper PRO 5975WX 32-core CPU, 503 GB of RAM, and three NVIDIA RTX A6000 GPUs (48 GB VRAM each) running CUDA 12.8.

#### 2.5.1. Approach 1: The Baseline Model

To derive the most appropriate deep learning model for Approach 1, we tried four deep learning models to evaluate their effectiveness in capturing temporal and contextual dependencies in the EEG data. This systematic evaluation of models provided insight into their strengths and weaknesses, ultimately leading to selecting the most suitable approach for emotion recognition. Specifically, we used a convolutional neural network (CNN) [36] to extract spatial features. Second, we used a Graph Neural Network (GNN) [37] to relate the DE features of the channels based on the proximity between them. Since the first two models focused more on spatial relationships, we also explored other models to capture time variations. Thus, in our third model, we equipped the GNN with a long short-term memory (LSTM) network [38] to improve sequential modeling. Finally, to further capture long-range dependencies, we used a model composed of a GNN and a transformer [39].

Each model (CNN, GNN, GNN + LSTM, and GNN + Transformer) was implemented with comparable representational capacity. The CNN consisted of two convolutional layers followed by a fully connected output layer. The first convolutional layer used 32 filters, 3 × 3 kernel, and a stride of 1, followed by max-pooling with a size of (2, 1) and a dropout of 0.2. The second convolutional layer used 64 filters, with the same kernel and pooling configuration. The output from the final convolution layer was flattened and passed through a fully connected layer that produced the three output classes.

The standalone GNN model consisted of a single graph convolutional layer using a normalized adjacency matrix to define node relationships with five input features and eight output features. A ReLu activation function is applied after this layer, followed by a dropout rate of 0.2. This is followed by a fully connected layer mapping to 3 output classes.

The GNN + LSTM model used the same GNN described above. The resulting spatial features are flattened and fed into a bidirectional LSTM with 32 hidden units to capture temporal dynamics across windows. The concatenated forward and backwards states were passed through a dropout and a fully connected layer.

In the GNN + Transformer model, each EEG window (62 nodes × 5 features) was first processed by the same GNN layer described previously, producing eight features per node with SiLU activation, dropout, and layer normalization. The resulting node embeddings were flattened into tokens of size 62 × 8 = 496 and combined with a learnable positional embedding. These tokens were passed through a single Transformer encoder layer with 4 attention heads, a feed-forward layer with a dimension of 992 (2 × 496), and a dropout rate of 0.3. The encoded sequence was mean-pooled, followed by dropout 0.3 and a fully connected layer that produced the three output classes. Table 3 summarizes the architectures of the used models.

#### 2.5.2. Approach 2: The Extended Model

Approach 2 extends Approach 1 by combining its output with demographic variables. Three techniques were explored to perform this task: concatenation, CNN fusion, and attention fusion [15]. Because attention fusion achieved the highest performance, we selected it for Approach 2, as shown in Figure 1.

The architecture was an extension of the GNN + Transformer model. Specifically, it extended that model by including demographics, such as age, gender, and nationality features after Transformer mean pooling. The architecture consisted of two branches: one processed 310 EEG-derived features using the same deep neural model from Approach 1, while the other encoded the demographic attributes, namely sex, age, and nationality. Each branch was passed through a fully connected layer, reducing the EEG output to 16 units ($O_{1}$) and expanding the demographic input from 4 to 16 units ($O_{2}$). The resulting embeddings ($O_{1}$ and $O_{2}$) were then fused using an attention mechanism. Specifically, each of them was multiplied element-wise with a shared 16-unit weight vector (*A*), and attention weights $\omega_{1}$ and $\omega_{2}$ were computed as: $\omega_{1}=\frac{\exp(A\cdot O_{1})}{\exp\left(A\cdot O_{1}\right)+\exp\left(A\cdot O_{2}\right)}$, and $\omega_{2}=1-\omega_{1}$. The final fused features were computed as $O_{fusion}=\omega_{1}O_{1}+\omega_{2}O_{2}$.

### 2.6. Hyperparameter Selection

To ensure a fair and unbiased comparison across all models and the various approaches, we conducted the same hyperparameter search strategy to identify the best hyperparameters. Specifically, each model underwent an identical grid search over a range of values for learning rate and weight decay, ensuring that no model had any preferential tuning. This strategy allows each architecture to be evaluated to near optimal configuration, allowing for an unbiased comparison across architectures.

The search included learning rate, dropout rate, and many other key parameters specific to each approach, which were tuned by evaluating performance on the training data and selecting the values that yielded the highest accuracy. While early stopping can prevent overfitting, we chose to train each model on a fixed number of epochs to maintain a consistent number of training epochs across all subjects for our leave-one-subject-out cross-validation (LOSOCV). This approach ensured consistency across subjects in the evaluation process while leveraging the individually optimized hyperparameters.

Table 4 summarizes the selected hyperparameters optimized. All training approaches used a batch size of 32. To select optimal learning rates and weight decay, we performed a grid search over values for the learning rate ($10^{-5},10^{-4},10^{-3},10^{-2}$) and weight decay ($0,10^{-4},10^{-3},10^{-2},10^{-1}$). We selected the configuration that produced the best overall model performance for each of the models tested. The simpler CNN and GNN models were trained using Stochastic Gradient Descent (SGD), whereas the larger models, GNN + LSTM and GNN + Transformer models, used the AdamW optimizer with heavy regularization using a weight decay of 0.1.

Importantly, tuning each model within the same search space ensures that the comparison reflects the models’ actual capabilities rather than arbitrary hyperparameter choice.

### 2.7. Performance Metrics

To evaluate model performance in a subject-independent manner, we employed again LOSOCV on model given by the best hyperparameters. This approach involved iterating over each of the 31 subjects, using the data from the selected subject as the test set while training the deep learning model on data from the remaining 30 subjects. Consequently, the LOSOCV process yielded a separate performance metric for each subject.

Since the datasets were balanced across the emotion classes, we used accuracy to evaluate the performance of the deep learning models. Accuracy was calculated in two ways: per-subject performance and per-emotion performance. Per-subject performance was computed separately for each subject to evaluate how well the model performed across different users. This approach accounted for potential variations due to demographic imbalances and provided insight into the model’s ability to generalize. Per-emotion accuracy was used to measure the model’s effectiveness in distinguishing between different emotional states, helping to identify whether certain emotions were more easily classified than others. Additionally, we computed the macro F1-score and the macro area under the receiver operating characteristic curve (ROC-AUC).

### 2.8. Impact of Demographics

To evaluate whether incorporating demographic information improved prediction accuracy, we calculated the performance difference for each subject between the two approaches. For each demographic variable, we compared performance differences across their respective categories: female and male for biological sex; Chinese, French, and German for nationality; and younger than or equal to 23 years versus older than 23 years for age. To determine whether these differences were statistically significant, we conducted a two-sided paired Wilcoxon signed-rank test. The null hypothesis posited that the performance difference between the baseline and extended approaches was symmetric around zero, while the alternative hypothesis assumed an asymmetric distribution of differences around zero. The hypothesis test provided *p*-values, where a lower *p*-value suggested statistically significant differences.

To avoid false positives due to multiple comparisons, we adjusted *p*-values using Bonferroni correction. For reference, we interpreted *p*-values between 0.1 and 0.05 as weak evidence, between 0.05 and 0.01 as moderate evidence, and less than 0.01 as strong evidence of a significant difference.

### 2.9. Demographic Ablation Analysis

To evaluate the contribution of demographic information to model performance, we conducted an ablation study in which each demographic variable was removed individually. The goal of this analysis was to identify which variable caused the greatest performance reduction when excluded. A significant decrease in performance indicates that the variable plays a crucial role in emotion prediction and should therefore be carefully considered when developing subject-independent emotion recognition models.

### 2.10. Demographic Feature Importance

To analyze the extent to which the extended model (Approach 2) relied on EEG-derived features and demographic information, we visualized the average weights ($\omega_{1}$ and $\omega_{2}$) obtained across the 31 subjects. This analysis allowed us to determine whether the extended model focused more on one of these two inputs or if both inputs contributed equally to the emotion predictions. Additionally, we measured the importance of the demographic variables by considering the expanded feature representation shown in Figure 1, which was defined as:(4)$$O_{2}=f\left(X_{2}\right)=X_{2}L,$$
where $X_{2}$ was the matrix containing the original features of dimension N×4, *L* was the matrix associated with the linear layer used for expansion, and $O_{2}$ was the expanded demographic data of dimension N×16. As the contribution of the expanded demographic was given by $\omega_{2}⊙O_{2}$, the contribution of for the *j*-th dimension of $O_{2}$ was:(5)$$\omega_{2,j}O_{2}^{\left(j\right)}=\omega_{2,j}X_{2}L^{\left(j\right)}=X_{2}\left(\omega_{2,j}L^{\left(j\right)}\right),$$
where $\omega_{2,j}$ was the *j*-th entry of the $\omega_{2}$ vector, and $L^{\left(j\right)}$ was the *j*-th column of the matrix *L*. Thus, the contribution of each original demographic features corresponded to the rows of the matrix $\omega_{2}⊙L$.

To aggregate the contributions of each feature, we first computed the absolute value of $\omega_{2}⊙L$. We then calculated the average across columns, resulting in a vector of four entries. These vectors were normalized per subject, and the mean across subjects was subsequently calculated to identify the demographic features that were most relevant across the cohort.

## 3. Results

### 3.1. Model Selection

Table 5 and Table 6 present the performance of the models evaluated under Approach 1. The CNN achieved the lowest performance, likely due to its limited ability to capture temporal dependencies and its assumption of a grid-like data structure. To address spatial relationships more effectively, a GNN was employed, which improved the results. Building on this, we combined the GNN with an LSTM to model sequential dependencies, leading to an overall accuracy of 80%. Finally, replacing the LSTM with a transformer further enhanced the ability to capture both temporal and spatial patterns, yielding a performance of 82%. Consequently, the GNN+Transformer architecture was selected as the baseline model.

### 3.2. Model Performance

Figure 2 illustrates the average training and testing accuracy for the extended model across 80 epochs. The model shows stable convergence, with training accuracy steadily increasing toward 100 percent. Test accuracy, however, plateaued around 84 percent after approximately 25 epochs, indicating a clear generalization gap of roughly 16 percent. Early stopping was employed during training, but it did not significantly change the observed accuracy trends. Importantly, the test curve remained stable and did not decrease, suggesting that although the model overfitted to some extent as expected for deep models trained on EEG data, it did not exhibit harmful divergence or collapse.

Table 7 presents the average, standard deviation, and 95% confidence intervals for the recall, precision, and macro F1-score across the 31 subjects. The extended model achieved a balanced performance across the three classes, as reflected by F1-scores ranging from 82.3% to 88.5%. To further examine the class-wise distribution, Figure 3 displays the confusion matrix obtained via LOSOCV. For the negative samples, 9.8% were misclassified as neutral, while 7.4% of the positive samples were classified as negative. Nevertheless, the diagonal values indicate that the model consistently achieved performance above 82% for all three classes.

### 3.3. Comparison Between Baseline and Extended Models

Figure 4 shows the difference, per subject, between the extended and baseline models in predicting negative, neutral, positive, and overall emotions. The overall accuracy of the extended model decreased for 15 subjects, increased for another 15, and showed no change for one subject. The extended model improved performance for negative and positive emotions by 3.37% and 7.33%, respectively. It also achieved an average overall improvement of 2.02% across all subjects.

Regarding overall improvement, Table 8 shows the 95% confidence intervals for the difference between the extended and baseline models after including demographic variables. For negative and positive emotions, the confidence intervals indicated significant improvements of 1.1–7.9% and 0.9–5.9%, respectively. These effects were reflected in the overall prediction, where the 95% CI for the improvement across the 31 subjects was 1.6–6.1%.

### 3.4. Performance by Demographic Group

Each subplot (a–d) in Figure 5 displays a specific emotion within each nationality group. Significant improvement was observed in the German group for negative emotions (two-sided Wilcoxon rank-sum test; *p*-value = 0.04). For neutral, the Chinese group showed a near-significant improvement (*p*-value = 0.07), while the French group showed significant improvement for positive emotions (*p*-value = 0.04). Overall, emotions showed suggestive improvements in the Chinese (*p*-value = 0.07) and German (*p*-value = 0.08) groups.

Figure 6 shows the performance difference between approaches by sex groups. We observed a near-significant improvement for males in recognizing negative emotions (*p*-value = 0.07). For neutral emotions, a near-significant improvement was observed for females (*p*-value = 0.08), while for positive emotions, a significant improvement was observed in the female group (*p*-value = 0.02). Overall, males showed a suggestive improvement in accuracy (*p*-value = 0.08), whereas females showed a significant improvement (*p*-value = 0.01).

Figure 7 shows the comparison for the age groups. For negative emotions, there was a significant improvement for ages younger than 23 (*p*-value = 0.05). For neutral, there were no significant improvements for both age groups (*p*-value = 0.35 and *p*-value = 0.23). Both groups resulted in a near-significant improvement in positive emotions. Finally, for the overall predictions, there was a weak improvement for those over 23 (*p*-value = 0.07), while those younger than 23 resulted in a significant improvement (*p*-value = 0.02).

### 3.5. Demographic Ablation Analysis Results

Table 9 shows the results of an ablation study for evaluating the performance of the extended model when each demographic variable was individually removed. Among the variables, nationality had the greatest impact, reducing the overall classification accuracy by 0.9%.

### 3.6. Fusion Weights

Figure 8 shows the learned attention weights assigned to each of the 16 fused dimensions for both EEG-derived features and demographic data. The variation in weights across dimensions indicates that the model does not treat all features equally during fusion. Notably, dimensions 9 and 11 exhibit contrasting patterns of modality dominance: dimension 9 assigns a low weight to EEG-derived features and a high weight to demographics, whereas dimension 11 assigns nearly equal weights (approximately 0.5) to both modalities. Other dimensions, such as dimension 15, demonstrate a stronger emphasis on EEG-derived features, with 60% of the weight assigned to this type. However, across the 16 dimensions, most provide roughly equal weighting to both feature types (between 45% and 55% for each), suggesting a balanced reliance on both feature types for emotion prediction.

### 3.7. Feature Importance

Figure 9 shows the attention weights assigned to each demographic variable. Among age, sex, and nationality, the model placed greater emphasis on the dummy-coded nationality variables (Chinese ‘00’, French ‘10’, German ‘01’), with weights exceeding 0.3 across all emotion categories. Among the two nationality variables, the first bit of the dummy variable received the highest overall weight, suggesting that the distinction between being French or not consistently played a prominent role in emotion classification. The variable sex was less important for predicting neutral emotions and had a greater influence on predicting positive emotions. Finally, the variable age showed higher influence for negative emotions and less for positive emotions.

## 4. Discussion

### 4.1. Impact of Demographics on Emotion Recognition

Our results indicate that demographic factors influence the prediction of emotions. This suggests that each demographic group processes emotion differently, and so the model identified different patterns based on the combination of demographic features. These differences are supported by the performed hypothesis, which resulted in significant improvements across the demographic group (see Figure 5, Figure 6 and Figure 7).

The predictive improvement obtained by the extended approach highlights the importance of incorporating demographic information into emotion recognition models. Including demographics not only enhances subject-independent approaches but also accounts for group-specific variations in emotional processing, leading to improved overall performance. Therefore, the demographics allow deep learning models to extract relevant patterns for each demographic group, thereby supporting the development of personalized and equitable emotion recognition systems.

Regarding demographics, nationality was the most influential variable in our predictions, as indicated by the attention-weight pattern (see Figure 9). The feature representing French participants (Nationality 1) received the highest attention weights across all emotions, suggesting that nationality plays a particularly significant role in emotion classification. This finding aligns with cross-cultural research indicating that emotional expression and perception are influenced by culturally specific display rules and norms, including variations in expressiveness [40,41,42,43]. Therefore, it is essential to include a cultural background component when conducting EEG-based emotion recognition studies. By considering cultural context, researchers can gain more accurate insights into how emotional expressions and perceptions differ across nationalities and avoid potential biases that may arise from ignoring these factors. This inclusion will enhance the robustness of emotion classification models and foster a deeper understanding of the interplay between culture and emotional processing.

### 4.2. Comparison with Other Works

Previous models also reported the effect of including demographics to improve subject-independent EEG-based emotion recognition. For instance, Li et al. [23] achieved a maximum improvement of 4.92% for valence and 6.25% for arousal compared to models without demographic input on the DEAP dataset [14]. Even greater gains were observed on the DREAMER dataset [44], with improvements of 8.94% and 7.24% for valence and arousal, respectively. While our model demonstrated a more modest 2.02% overall increase in classification accuracy, it was evaluated on more demographically and culturally diverse datasets (SEED, SEED-FRA, SEED-GER), supporting its stronger generalizability across populations. Moreover, unlike the concatenation-based fusion used in [23], our model employs an attention-based mechanism that dynamically weights EEG and demographic inputs. This approach not only improves interpretability but also reveals that different characteristics contribute unequally across dimensions, highlighting the nuanced role of demographic context in emotional processing. This adaptive fusion strategy provides a more flexible and fine-grained integration of modalities.

Other works, such as Peng et al. and Liu et al., have emphasized the impact of sex and nationality on EEG patterns, often relying on demographic-specific models [24,25]. While effective, such methods require separate training for each group. In contrast, our model handles multiple demographic factors within a single architecture, improving scalability and fairness. Our results suggest that incorporating nationality alongside age and sex not only improves model performance, but also promotes a more inclusive and representative understanding of emotional processing across populations. Finally, our findings align with the call for ethical and inclusive AI in Sheoran & Valderrama, reinforcing the value of demographic-aware emotion recognition models [26]. Finally, Table 10 compares models on the same datasets and shows that incorporating all three demographics yields performance comparable to or better than models using only a single demographic factor.

### 4.3. Limitations

One of the primary limitations in this study is the scarcity of publicly available EEG datasets. Existing datasets are not only limited in number but also relatively small in size. This poses a challenge when using Transformer-based architecture, which typically requires large amounts of data to effectively learn complex patterns and long-range dependencies. As a result, the limited training data likely contributed to reduced model accuracy and generalizability.

Another significant limitation pertains to the demographic information available in the dataset. For instance, all participants in the datasets fall within a narrow age range of 19 to 29 years. This 10-year span is insufficient to capture meaningful differences in how emotional processing may vary across distinct developmental stages, such as adolescence, adulthood, and seniors. Furthermore, the dataset includes only three nationalities which introduce a demographic bias. Consequently, the model might underperform when applied to individuals from underrepresented or unrepresented national backgrounds.

These limitations in demographic representation raise important ethical considerations. A key objective of incorporating demographic features into subject-independent models is to enhance their fairness and applicability across diverse populations. However, if the training data lacks diversity, the model may inherit and even amplify existing biases. This can compromise the model’s reliability when deployed in real-world settings involving more varied demographic groups. It is crucial to ensure that such models do not inadvertently marginalize individuals outside the demographic scope of the training data.

Finally, we note that we used a single controlled backbone (GNN + Transformer) for both baseline and demographic-extended models to isolate the effect of demographic covariates on interpretability and performance; comparison with large pretrained/foundation encoders (e.g., wav2vec2.0, HuBERT, or Transformer-only EEG models) is left to future work because such models require large-scale pretraining/fine-tuning and complicate interpretability analyses.

### 4.4. Future Work

To improve the model’s performance and generalization, future work could explore data augmentation techniques [55]. Since collecting large-scale, demographically diverse EEG datasets is challenging, data augmentation can increase the effective training sample size and introduce variability to enhance model robustness. For example, adding low-amplitude Gaussian noise can simulate natural variability in EEG recordings without distorting key patterns [55]. Time-window slicing, already used in our pipeline, can be expanded by treating it as a flexible, tunable process. Varying the window duration, adjusting the overlap rate, or even introducing randomized start times can create diverse temporal views of each trial while preserving the underlying emotional dynamics. Other augmentation techniques, such as temporal jittering (slightly shifting signals in time) or frequency-domain transformations that modify power in specific EEG bands (e.g., alpha, beta), can simulate subject-specific variations in brain activity. Future work could explore and evaluate combinations of these techniques to improve model performance, especially in the context of subject-independent emotion recognition.

Addressing the limited demographic diversity in the SEED datasets is an important direction for future research. Collaborations to develop or access more diverse datasets, along with the use of demographic-aware evaluation metrics, could enhance model fairness and generalizability. Specifically, future studies will aim to include participants from broader age ranges, additional nationalities and cultural backgrounds, and balanced sex distributions to reduce demographic bias. In the short term, techniques such as synthetic data generation and domain adaptation may help investigate the impact of demographic variability on model performance. Cross-dataset validation will also be explored to assess the model’s robustness and generalizability across different EEG datasets.

Finally, future work may investigate alternative or hybrid deep learning architectures and transfer learning from larger EEG datasets to improve model generalization and capture complex EEG patterns, even in scenarios with limited training data. Together, these approaches aim to enhance the practical applicability, fairness, and robustness of subject-independent EEG-based emotion recognition models.

## 5. Conclusions

This study presents a deep learning approach that combines EEG signals with demographic information to improve emotion recognition in subject-independent scenarios. To that end, we developed and tested our model using three benchmark datasets: SEED, SEED-FRA, and SEED-GER. By incorporating three demographic variables, nationality, biological sex, and age, the predictions of emotions were significantly enhanced. These findings suggest that including demographic data can improve emotion recognition across different individuals, leading to fairer and more accurate results in diverse populations. Therefore, this underscores the importance of demographic-aware modeling in developing personalized and equitable emotion recognition systems.

## Figures and Tables

**Figure 1 diagnostics-16-00144-f001:**
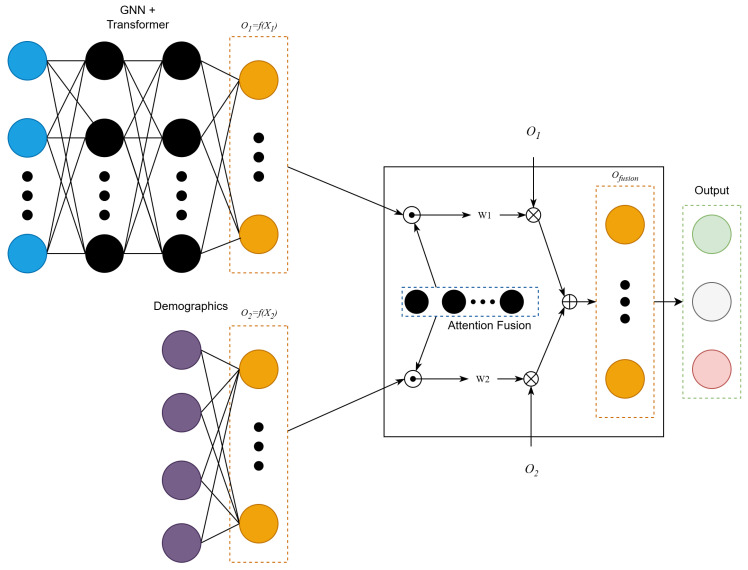
The GNN and Transformer reduce 310 EEG-derived DE features to a 16-dimensional vector ($O_{1}$). Demographic features (age, sex, nationality) are projected from 4 to 16 dimensions ($O_{2}$). An attention module learns weights $\omega_{1}$ and $\omega_{2}$ to fuse $O_{1}$ and $O_{2}$ before classification.

**Figure 2 diagnostics-16-00144-f002:**
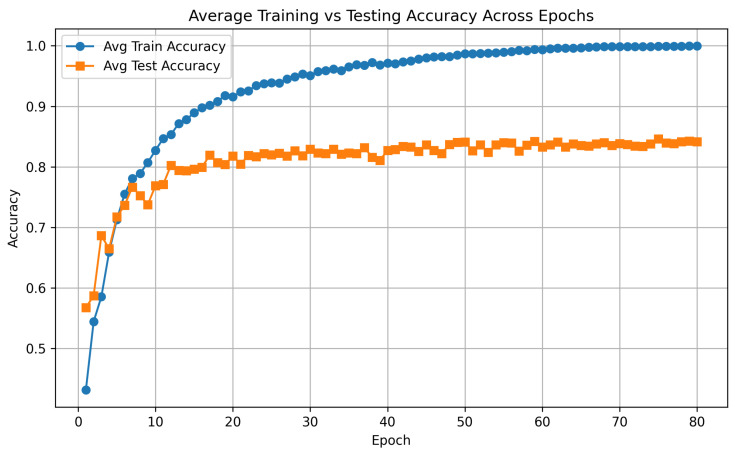
Training and testing accuracy averaged over all subjects per-epoch across 80 epochs.

**Figure 3 diagnostics-16-00144-f003:**
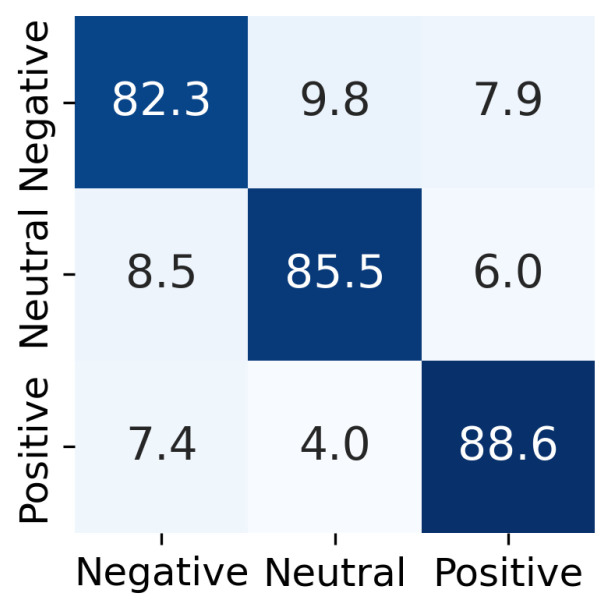
Confusion matrix for the extended model.

**Figure 4 diagnostics-16-00144-f004:**
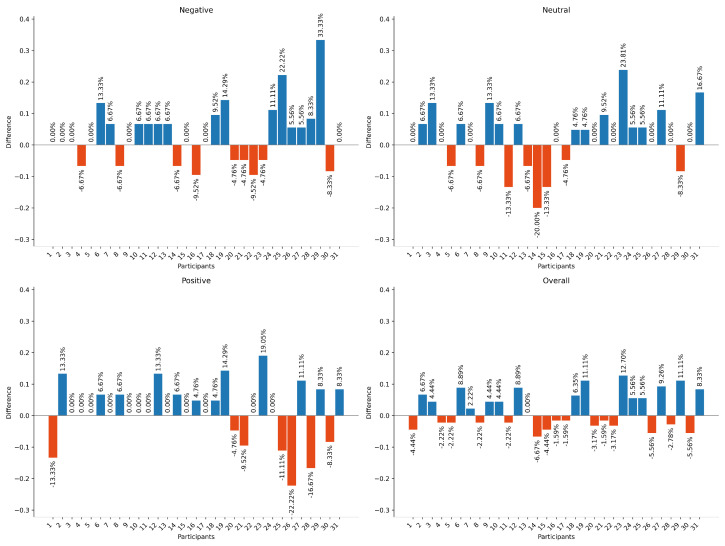
Performance differences between the extended and baseline models in predicting negative, neutral, positive, and overall emotions. Blue bars represent improvements in prediction accuracy after incorporating demographic information, while red bars indicate a reduction in performance.

**Figure 5 diagnostics-16-00144-f005:**
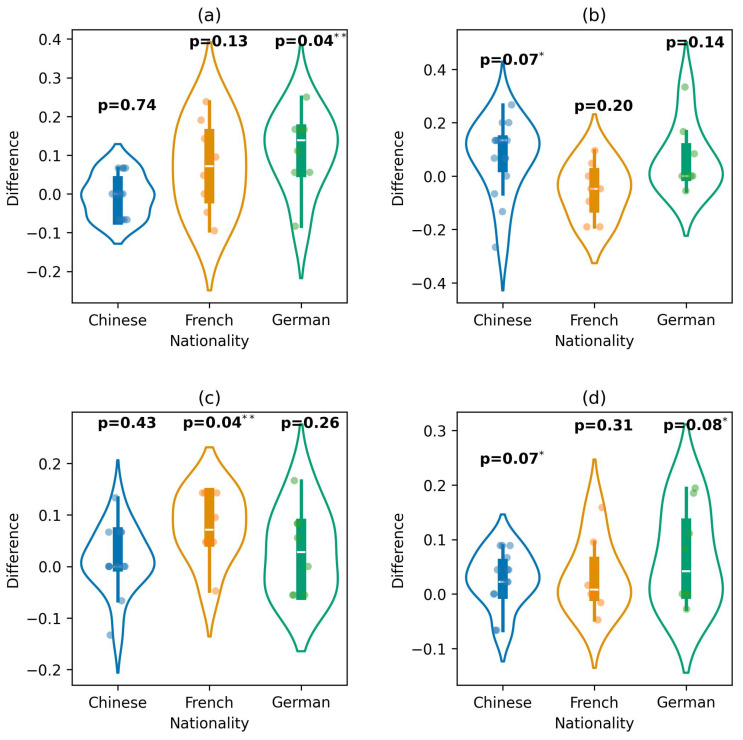
Performance difference between the extended and baseline models across nationality groups for predicting: (**a**) negative, (**b**) neutral, (**c**) positive, and (**d**) overall emotions (* *p*-value < 0.1; ** *p*-value < 0.05).

**Figure 6 diagnostics-16-00144-f006:**
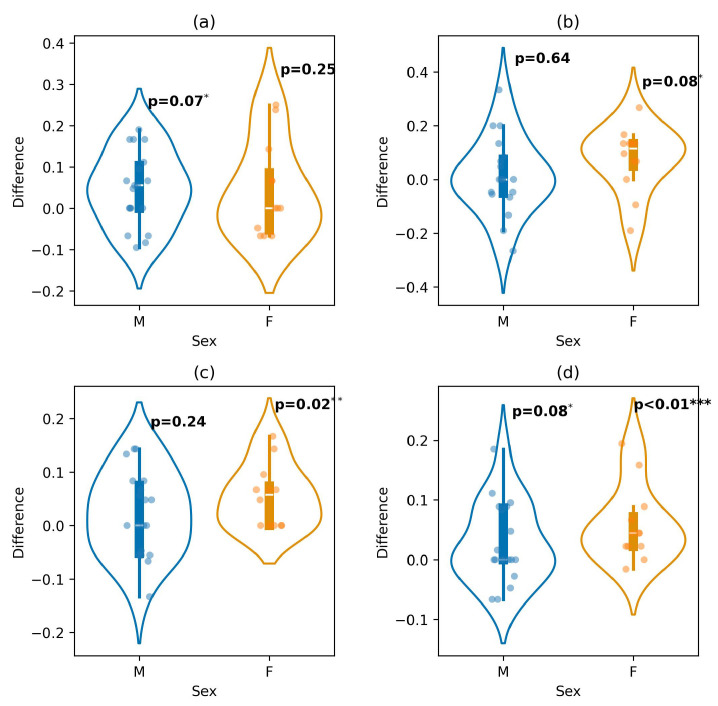
Performance difference between the extended and baseline models across sex groups for predicting: (**a**) negative, (**b**) neutral, (**c**) positive, and (**d**) overall emotions (* *p*-value < 0.1; ** *p*-value < 0.05; *** *p*-value < 0.01).

**Figure 7 diagnostics-16-00144-f007:**
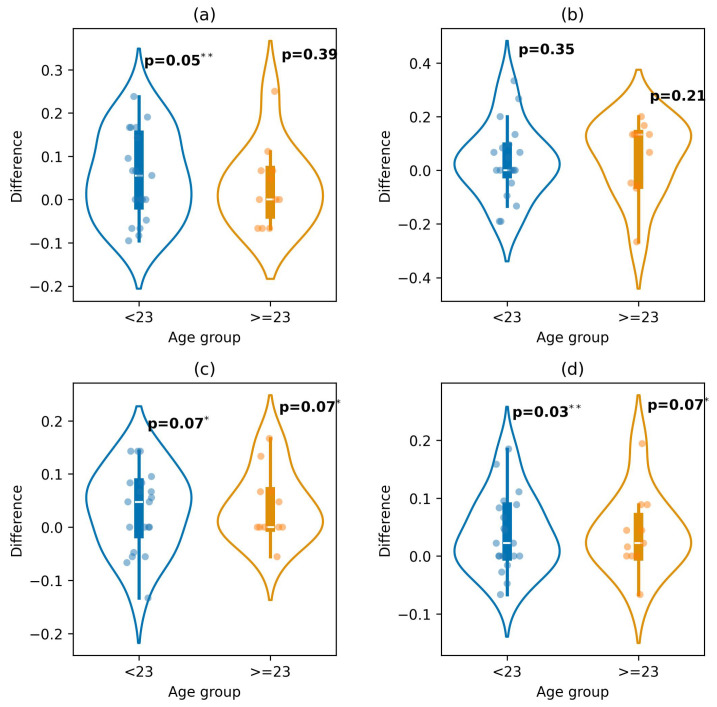
Performance difference between the extended and baseline models across age groups for predicting: (**a**) negative, (**b**) neutral, (**c**) positive, and (**d**) overall emotions (* *p*-value < 0.1; ** *p*-value < 0.05).

**Figure 8 diagnostics-16-00144-f008:**
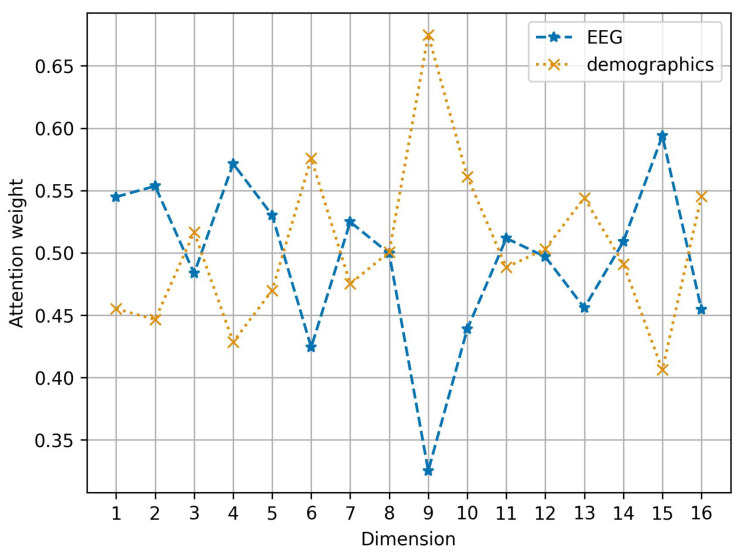
Average weights ($\omega_{1}$ for EEG features and $\omega_{2}$ for demographic features) for the 16 dimension of the attention fusing model.

**Figure 9 diagnostics-16-00144-f009:**
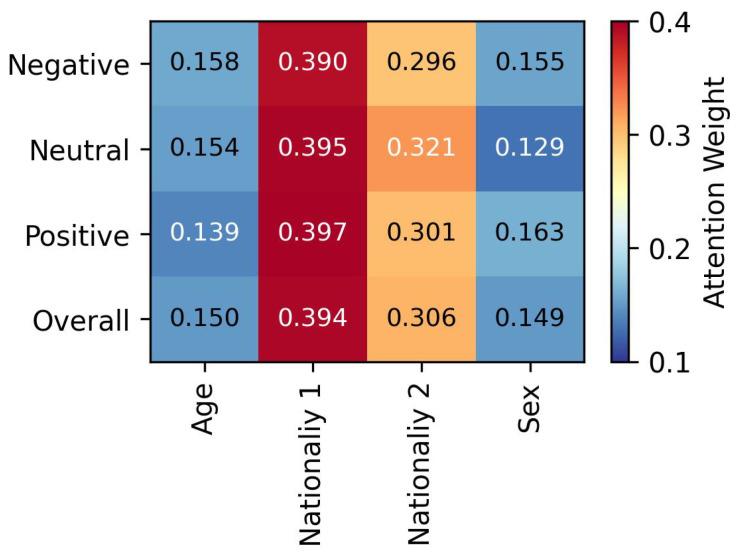
Average attention weights (scaled by $\omega_{2}$) assigned to the four demographic variables across emotion categories. Higher values (red tones) indicate greater relative importance of the demographic feature in emotion prediction.

**Table 1 diagnostics-16-00144-t001:** Summary of related work on demographic-aware EEG-based emotion recognition.

Study	Dataset/Subjects	Demographic Factors	Method/Experimental Design	Key Findings
Li et al. (2022) [23]	SEED (same nationality)	Age, Sex	Demographic variables appended at final layer of deep network under subject-independent setting	Age + sex improved recognition performance; cultural factors not considered
Peng et al. (2023) [24]	SEED (Chinese subjects)	Sex	Compared same-sex vs. cross-sex training/testing workflows	Same-sex models outperformed cross-sex models; demographic mismatch degrades performance
Liu et al. (2022) [25]	SEED, SEED-GER, SEED-FRA	Nationality	Compared within-nationality vs. cross-nationality deep learning pipelines	Models trained/tested on same nationality achieved highest accuracy; cultural background affects emotion-related EEG patterns
Sheoran et al. (2025a) [26]	SEED-family	Sex, Age, Nationality	Demographic-informed model using auxiliary metadata fusion	Including sex, age, nationality increased likelihood of correct prediction
Sheoran et al. (2025b) [27]	SEED-family	Sex, Age, Nationality	Evaluated demographic-dependent architectures under subject-independent setup	Biological sex and nationality strongly affect model generalization

**Table 2 diagnostics-16-00144-t002:** Descriptions of the SEED, SEED-FRA, and SEED-GER datasets. For each dataset, the total number of subjects, number of EEG recordings per subject, nationality, male/female ratio, and average subject age are provided.

Dataset	Subjects	EEG per Subject	Nationality	Male/Female	Average Age
SEED [34]	15	45	Chinese	7/8	23.27
SEED-FRA [25]	8	63	French	5/3	22.50
SEED-GER [25]	8	54	German	7/1	22.25

**Table 3 diagnostics-16-00144-t003:** Model architecture summary. Abbrev.: B = batch, W = windows, N = 62 nodes, F = 5 features, g = graph feat dim, h = LSTM hidden.

Model	Layers/Modules	Input → Output	Key Dims
CNN	2 × Conv2D + MaxPool + FC	(B,1,N,F)→(B,3)	32/64 filters; pool 2 × 1; drop 0.2
GNN	GraphConv + FC	(B,N,F)→(B,3)	g=8; ReLU + LayerNorm; drop 0.3
GNN + LSTM	GraphConv + BiLSTM + FC	(B,W,N,F)→(B,3)	g=8; h=32 (bi, concat); drop 0.3
GNN + Transf. (no demo)	GraphConv + Transformer + FC	(B,W,N,F)→(B,3)	g=8; heads = H; 1 layer
GNN + Transf. (demo)	GraphConv + Transformer + Fusion + FC	(B,W,N,F) + (B,3)→(B,3)	g=8; heads = H; 1 layer; fusion: Attn(16)

**Table 4 diagnostics-16-00144-t004:** Hyperparameters for the CNN, GNN, GNN + LSTM, and GNN + Transformer.

Model	Epochs	Learning Rate	Optimizer	Weight Decay
CNN	30	0.001	SGD	0
GNN	50	0.01	SGD	0
GNN + LSTM	80	0.001	AdamW	0.1
GNN + Transformer	80	0.001	AdamW	0.1

**Table 5 diagnostics-16-00144-t005:** Performance per class of the CNN, GNN, GNN + LSTM, and GNN + Transformer for Approach 1.

Model	Negative (%)	Neutral (%)	Positive (%)
CNN	59	51	69
GNN	78	53	52
GNN + LSTM	73	87	80
GNN + Transformer	79	80	86

**Table 6 diagnostics-16-00144-t006:** Overall Performance of the CNN, GNN, GNN + LSTM, and GNN + Transformer for Approach 1.

Model	Accuracy (%)	Macro F1-Score (%)	Macro AUC (%)
CNN	60	62	80
GNN	61	63	81
GNN + LSTM	80	80	91
GNN + Transformer	82	82	93

**Table 7 diagnostics-16-00144-t007:** Recall, Precision, and F1-score macro for the extended model. For each metric, the mean, standard deviation (SD), and 95% confidence interval (CI) is provided.

Metric	Recall (%)	Precision (%)	Macro F1-Score (%)
Mean	85.5	86.2	85.4
SD	8.3	8.0	8.4
Lower CI	82.4	83.3	82.3
Upper CI	88.5	89.2	88.5

**Table 8 diagnostics-16-00144-t008:** 95% confidence intervals for the difference between the extended and baseline models for negative, neutral, positive, and overall emotion prediction. * indicates *p* < 0.05.

Type	Lower CI (%)	Upper CI (%)	*p*-Value
**Negative**	1.1	7.9	0.033 *
**Neutral**	−0.9	8.4	0.103
**Positive**	0.9	5.9	0.013 *
**Overall**	1.6	6.1	0.004 *

**Table 9 diagnostics-16-00144-t009:** Ablation study assessing the impact of removing each demographic variable from the extended approach. The Receiver Operating Characteristic Area Under the Curve (ROC-AUC) is reported for each emotion. The minus are used to indicate that the feature was not included.

Model	Negative (%)	Neutral (%)	Positive (%)	Overall (%)
All features	92.2	94	95.7	94.0
- age	93.1	94.5	95.1	94.3
- nationality	91.7	93.7	93.8	93.1
- sex	92.4	94.4	95.6	94.1

**Table 10 diagnostics-16-00144-t010:** Concise comparison of prior methods and our approach on SEED, SEED-FRA, and SEED-GER datasets. Performance reported as accuracy (Acc.) and standard deviation (SD).

Demographic Variable	Reference	Dataset	Performance (Acc./SD)
Nationality	KNN [25]	SEED	54.1/8.7
		SEED-FRA	37.2/6.8
		SEED-GER	41.0/7.4
	SVM [25]	SEED	72.6/10.5
		SEED-FRA	50.1/10.3
		SEED-GER	55.6/12.2
	LR [25]	SEED	68.4/11.7
		SEED-FRA	47.2/12.2
		SEED-GER	50.4/10.9
	DNN [25]	SEED	82.8/7.5
		SEED-FRA	64.2/8.6
		SEED-GER	65.9/10.1
	DL-LR [27]	SEED	77.2/5.3
		SEED-FRA	73.0/5.0
		SEED-GER	65.6/6.0
Sex	SVM [45]	SEED	83.2/9.6
	BiDANN [46]		83.2/9.6
	DGCNN [47]		79.9/9.0
	A-LSTM [48]		72.1/10.8
	IAG [49]		86.3/6.9
	RGNN [50]		85.3/6.7
	BiHDM [51]		85.4/7.5
	GECNN [52]		82.4/-
	BiHDM w/o DA [53]		81.5/9.7
	GMSS [53]		86.5/6.2
	JD-IRT [54]		83.2/-
	Graph-LSTM [8]		79.3/5.8
	DL-LR [26]		81.5/7.8
Sex, Age, Nationality	Ours	SEED	85.2/5.6
		SEED-FRA	78.2/7.4
		SEED-GER	72.1/6.6

## Data Availability

The data presented in this study are available in the SEED project website at https://bcmi.sjtu.edu.cn/home/seed/ (accessed on 14 March 2025), reference number 16.

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
