# Peer review of "Evaluating the Impact of Demographic Factors on Subject-Independent EEG-Based Emotion Recognition Approaches"

_diagnostics, 2026, doi:10.3390/diagnostics16010144_

Round 1
Reviewer 1 Report
Comments and Suggestions for Authors
Dear Authors,
Thank you for submitting the article. Please find below the comments:
- Add Literature Review Section and Literature Summary Table. Clearly state the research Gap, and then outline your major contributions.
- Refer to all equations, figures, and tables in the text also.
- Model details, such as layers and neurons, are not mentioned for CNN, GNN, GNN + LSTM, and GNN + Transformer. Additionally, the paper lacks details on other hyperparameters (serious concern).
- Add an abstract level diagram of the study.
- Add hardware setup details.
- Details such as Batch Size, Regularization, and Loss functions are not available in the paper (serious concern).
- Add learning graphs of the model training (serious concern).
- Have you considered the early stopping?
- Is the learning rate 0.001 throughout the training? If so, then it is better to consider the reduced learning rate method.
- Add practical implications of the study.
- Perform the comparison with state-of-the-art studies in tabular form.
Author Response
- Add Literature Review Section and Literature Summary Table. Clearly state the research Gap and then outline your major contributions.
>> We thank the reviewer for your suggestion. In the new version, we have reviewed and edited the introduction. Our contribution was included in the introduction to read as:
“Previous studies have demonstrated the potential of demographic information for improving the performance of subject-independent emotion recognition approaches. Incorporating factors such as sex, age, or nationality can enable the extraction of group-specific patterns that enhance prediction accuracy. However, existing works present two main limitations. First, Li et al. [13] only considered age and sex within a dataset of subjects from the same nationality, thereby neglecting the influence of cultural background on emotion perception, a factor that previous research has identified as critical when comparing interdependent and individualistic societies [18,19]. Second, the comparisons made in [14,15] between same-sex and same-nationality models and their cross-sex and cross-nationality counterparts limit practical applicability, as demographic group membership may not always be known in advance. Moreover, this strategy requires training and maintaining multiple deep learning models tailored to each demographic subgroup, which is impractical in scenarios with limited data availability for specific populations. Therefore, further research is needed to develop a unified approach that integrates demographic information with EEG signals, enabling models to generalize across diverse populations irrespective of sex, age, or cultural background.
To address these limitations, the present study proposes a deep learning framework that fuses EEG signals with demographic information to improve emotion recognition under a subject-independent setting. Specifically, we evaluate our approach using three benchmark emotion recognition datasets (SEED, SEED-FRA, and SEED-GER), which include EEG data from 31 participants of different sexes, ages, and nationalities. The main contributions of this work are as follows:
- we propose a unified deep learning framework that integrates EEG features with three key demographic variables (sex, age, and nationality) for emotion recognition
- we analyze the relative influence of each demographic variable on recognition performance, providing insights into their individual and combined effects
- we show that fusing demographic information enhances the generalization capability of subject-independent emotion recognition models across diverse populations.”
- Refer to all equations, figures, and tables in the text also.
>> We are sorry for this confusion. We have checked the whole document, and we have referred to all equations, figures, and tables in the text also.
- Model details, such as layers and neurons, are not mentioned for CNN, GNN, GNN + LSTM, and GNN + Transformer. Additionally, the paper lacks details on other hyperparameters (serious concern).
>> Thank you for your suggestion. We revised our methodology to include detailed model architecture and hyperparameter selection for all models. Please refer to section 2.5.1 and 2.5.2. Our model details can be found below:
“Each model (CNN, GNN, GNN + LSTM, and GNN + Transformer) was implemented with comparable representational capacity. The CNN consisted of two convolutional layers followed by a fully connected output layer. The first convolutional layer used 32 filters, 3x3 kernel, and a stride of 1, followed by max-pooling with size of (2, 1) and dropout of 0.2. The second convolutional layer used 64 filters, with the same kernel and pooling configuration. The output from the final convolution layer was flattened and passed through a fully connected layer that produced the 3 output classes.
The standalone GNN model consisted of a single graph convolutional layer using a normalized adjacency matrix to define node relationships with 5 input features and 8 output features. A ReLu activation function is applied after this layer, followed by a dropout rate of 0.2. This is followed by a fully connected layer mapping to 3 output classes.
The GNN + LSTM model used the same GNN described above. The resulting spatial features are flattened and fed into a bidirectional LSTM with 32 hidden units to capture temporal dynamics across windows. The concatenated forwards and backwards states were passed through dropout and a fully connected layer.
In the GNN + Transformer model, each EEG window (62 nodes x 5 features) was first processed by the same GNN layer described previously, producing eight features per node with SiLU activation, dropout, and layer normalization. The resulting node embeddings were flattened into tokens of size 62x8 = 496 and combined with a learnable positional embedding. These tokens were passed through a single Transformer encoder layer with 4 attention heads, a feed-forward with dimension of 992 (2 x 496), and a dropout rate of 0.3. The encoded sequence was mean-pooled, followed by dropout 0.3 and a fully connected layer that produced the three output classes.”
- Add an abstract level diagram of the study.
>> We thank the reviewer for the suggestion. As our current manuscript is lengthy, we have included only the diagram of the final model in Figure 1. The description of the architecture was included as “The architecture consisted of two branches: one processed 310 EEG-derived features using the same deep neural model from Approach 1, while the other encoded the demographic attributes—sex, age, and nationality. Each branch was passed through a fully connected layer, reducing the EEG output to 16 units (O₁) and expanding the demographic input from 4 to 16 units (O₂). The resulting embeddings (O₁ and O₂) were then fused using an attention mechanism. Specifically, each of them was multiplied element-wise with a shared 16-unit weight vector (A), and attention weights ω₁ and ω₂ were computed as: ω₁ = exp(A·O₁) / [exp(A·O₁) + exp(A·O₂)], and ω₂ = 1 − ω₁.The final fused features were then computed as: O_fusion = ω₁·O₁ + ω₂·O₂”
- Add hardware setup details.
>> We appreciate this suggestion. In the new version, we have included the technical details of the hardware setup used for our experiments in the revised manuscript: CPU – AMD Ryzen Threadripper PRO 5975WX (32 cores), RAM – 503 GB, and GPU – 3 × NVIDIA RTX A6000 (48 GB VRAM each, CUDA 12.8).
- Details such as Batch Size, Regularization, and Loss functions are not available in the paper (serious concern).
>> Thank you for your suggestion. We revised our methodology to include subsection 2.6 hyperparameter selection. This section goes into detail of the final hyperparameters used for each model, including learning rate, epochs, optimizer and weight decay. A table is provided to summarize this information. A grid search was performed on a range of values for learning rates and weight decay. Our new section reflects this information and reads as follows:
“To ensure optimal model performance across the different approaches, we conducted separate searches for the best hyperparameters. The search included parameters such as learning rate, dropout rate, and other key settings specific to each model, which were tuned by evaluating performance on the training data and selecting the values that yielded the highest accuracy. While early stopping can help prevent overfitting, we chose to train each model for a fixed number of epochs to maintain consistency across all subjects during our leave-one-subject-out cross-validation (LOSOCV). This ensured uniform evaluation conditions while still benefiting from individually optimized hyperparameters.
Table 3 summarizes the selected hyperparameters. All training approaches used a batch size of 32. To determine the optimal learning rates and weight decay values, we performed a grid search over the following ranges: learning rate {10⁻⁵, 10⁻⁴, 10⁻³, 10⁻²} and weight decay {0, 10⁻⁴, 10⁻³, 10⁻², 10⁻¹}. The configuration that achieved the best overall performance was selected for each model. The simpler CNN and GNN models were trained using Stochastic Gradient Descent (SGD), while the more complex GNN+LSTM and GNN+Transformer models used the AdamW optimizer with strong regularization (weight decay = 0.1).”
- Add learning graphs of the model training (serious concern).
>> We appreciate the reviewer’s suggestion to include visual performance comparisons. However, because the manuscript already contains many figures and we trained several models, adding a dedicated performance figure for each model would significantly increase the total number of figures and may reduce overall readability. To address this concern while keeping the presentation concise, we have included two summary tables that report on the performance of all models. Table 1 presents the per-class results (Negative, Neutral, Positive), and Table 2 provides the overall performance metrics (Accuracy, Macro F1-score, and Macro AUC). These tables offer a clear and compact comparison without introducing additional figures.
Table 1. Performance per class of the CNN, GNN, GNN + LSTM, and GNN + Transformer for Approach 1.
|
Model |
Negative (%) |
Neutral (%) |
Positive (%) |
|
CNN |
59 |
51 |
69 |
|
GNN |
78 |
53 |
52 |
|
GNN + LSTM |
73 |
87 |
80 |
|
GNN + Transformer |
79 |
80 |
86 |
Table 2. Overall performance of the CNN, GNN, GNN + LSTM, and GNN + Transformer for Approach 1.
|
Model |
Accuracy (%) |
Macro F1-score (%) |
Macro AUC (%) |
|
CNN |
60 |
62 |
80 |
|
GNN |
61 |
63 |
81 |
|
GNN + LSTM |
80 |
80 |
91 |
|
GNN + Transformer |
82 |
82 |
93 |
- Have you considered the early stopping?
>> We did consider early stopping but decided to opt out for the following reasons. First, we did initially explore early stopping but did not find improvements through initial experimentation. Second, all the models trained used a fixed number of epochs, and the evaluation metrics correspond to the final trained model at the last epoch. We opted for this approach to maintain consistency across subjects in our Leave-One-Subject-Out-Cross-Validation (LOSOCV) evaluation and to reduce the number of additional hyperparameter decisions. Since each subject can have different learning dynamics and data distributions, applying early stopping individually could introduce variability in training duration and potentially bias the aggregated results. Early stopping can help prevent overfitting, so we prioritized consistency across subjects and models. Section 2.5, “Hyperparameter Selection’, was added to describe the number of epochs used for training each model with a table that also summarizes the hyperparameter selection.
- Is the learning rate 0.001 throughout the training? If so, then it is better to consider the reduced learning rate method.
>> Thank you for the suggestion. We did not use a fixed learning rate of 0.001 for all models. Instead, the optimal learning rate was determined individually for each model through a grid search procedure, as described in Section 2.5, “Hyperparameter Selection’. The final selected values are summarized in the corresponding hyperparameter table. This approach allowed us to identify the most effective learning rate for each model configuration rather than applying a single constant value across all experiments.
- Add practical implications of the study.
>> Thank you for the suggestion. We have reviewed the discussion to include an implication of our work. Specifically, we indicate that “it is essential to include a cultural background component when conducting EEG-based emotion recognition studies. By considering cultural context, researchers can gain more accurate insights into how emotional expressions and perceptions differ across nationalities and avoid potential biases that may arise from ignoring these factors. This inclusion will enhance the robustness of emotion classification models and foster a deeper understanding of the interplay between culture and emotional processing.”
- Perform the comparison with state-of-the-art studies in tabular form
>> We appreciate the reviewer’s suggestion to include a tabular comparison with state-of-the-art studies. However, we believe that such a table would not be fully representative in this case, as the existing studies differ substantially in datasets, preprocessing pipelines, and evaluation settings (e.g., subject-dependent vs. subject-independent). To address this point more meaningfully, we have instead expanded the literature review in the Introduction to provide a broader contextual comparison and to highlight how our approach differs from prior work in terms of model design and experimental setup.
Reviewer 2 Report
Comments and Suggestions for Authors
The manuscript entitled “Evaluating the Impact of Demographic Factors on Subject-Independent EEG-Based Emotion Recognition Approaches” presents a deep learning framework that fuses EEG-derived features with demographic information (age, sex, nationality) through an attention-based mechanism. The study is timely and relevant to the field of EEG-based emotion recognition, addressing the important challenge of subject-independent generalization. While the work has potential, several issues must be addressed before it can be considered for publication.
- While the integration of demographic features into EEG-based emotion recognition is interesting, the manuscript does not sufficiently highlight how this approach differs from or improves upon existing demographic-aware or domain-adaptation methods. The authors should more clearly articulate the incremental contribution and practical novelty of their framework.
- The evaluation relies on three datasets (SEED, SEED-FRA, SEED-GER), which together provide only 31 subjects within a narrow age range (19–29 years) and limited nationalities (Chinese, French, German). These limitations restrict the generalizability of the conclusions. The authors should discuss these constraints more explicitly and outline how future work might address them, e.g., cross-dataset validation or inclusion of more diverse populations.
- The reported improvements (≈2% overall accuracy) appear modest. Although the authors use Wilcoxon signed-rank tests, more detailed statistical analyses are needed, such as reporting confidence intervals, effect sizes, and variability across subjects. This would help establish the reliability and significance of the observed gains.
- The attention-weight analysis suggests that nationality is the most influential demographic variable, but the interpretation remains superficial. The authors should provide deeper discussion linking these findings to psychological or cross-cultural theories of emotion perception. Case studies or illustrative examples would strengthen the interpretability claims.
- The comparative experiments are limited to CNN, GNN, GNN+LSTM, and GNN+Transformer. Given recent progress, it is important to compare against stronger baselines such as Transformer-based models, pretrained EEG/audio encoders (e.g., wav2vec2.0, HuBERT), or other foundation models for EEG. Without such comparisons, the competitiveness of the proposed method is unclear.
- While generally clear, the manuscript would benefit from a more concise and structured presentation. Some sections (e.g., model architecture and attention mechanism) are overly verbose and contain redundant explanations. The authors should simplify terminology, streamline descriptions, and consider adding a summary table or schematic to enhance readability.
- The introduction would benefit from citing additional recent and influential works in emotion recognition and multimodal/artificial intelligence contexts. These works provide state-of-the-art insights into spatiotemporal fusion for emotion recognition and offer a comprehensive review of AI-based emotion recognition research trends. In particular:[1]Hu et al., "STRFLNet: Spatio-Temporal Representation Fusion Learning Network for EEG-Based Emotion Recognition," IEEE Transactions on Affective Computing, 2025.[2]Hu, K. He, M. Qian, et al., "STAFNet: an adaptive multi-feature learning network via spatiotemporal fusion for EEG-based emotion recognition," Frontiers in Neuroscience, 2024.[3]Hazmoune S, Bougamouza F. Using transformers for multimodal emotion recognition: Taxonomies and state of the art review[J]. Engineering Applications of Artificial Intelligence, 2024, 133: 108339.
N/A
Author Response
The manuscript entitled “Evaluating the Impact of Demographic Factors on Subject-Independent EEG-Based Emotion Recognition Approaches” presents a deep learning framework that fuses EEG-derived features with demographic information (age, sex, nationality) through an attention-based mechanism. The study is timely and relevant to the field of EEG-based emotion recognition, addressing the important challenge of subject-independent generalization. While the work has potential, several issues must be addressed before it can be considered for publication.
- While the integration of demographic features into EEG-based emotion recognition is interesting, the manuscript does not sufficiently highlight how this approach differs from or improves upon existing demographic-aware or domain-adaptation methods. The authors should more clearly articulate the incremental contribution and practical novelty of their framework.
>> We thank the reviewer for this valuable comment and the opportunity to clarify the novelty and contribution of our proposed framework. We have revised the introduction section to more explicitly highlight how our approach differs from existing demographic-aware and domain-adaptation methods.
Specifically, prior studies have typically incorporated demographic factors such as sex or age by (1) training separate models for each demographic subgroup or (2) simply concatenating demographic variables with EEG features without deeper integration. These strategies limit generalization across diverse populations and require maintaining multiple models, which is impractical in real-world applications.
In contrast, our work introduces a unified deep learning framework that jointly learns from EEG signals and demographic embeddings within a single architecture. This design enables the model to leverage demographic cues during training while maintaining cross-demographic generalization at inference time, even when demographic labels are unavailable. Moreover, we provide a detailed analysis of the relative contribution of sex, age, and nationality to prediction performance, an aspect rarely explored in previous work.
The revised text in the introduction now reads:
“Previous studies have demonstrated the potential of demographic information for improving the performance of subject-independent emotion recognition approaches. Incorporating factors such as sex, age, or nationality can enable the extraction of group-specific patterns that enhance prediction accuracy. However, existing works present two main limitations. First, Li et al. [7] only considered age and sex within a dataset of subjects from the same nationality, thereby neglecting the influence of cultural background on emotion perception, a factor that previous research has identified as critical when comparing interdependent and individualistic societies [ 11 ,12]. Second, the comparisons made in [8, 9] between same-sex and same-nationality models and their cross-sex and cross-nationality counterparts limit practical applicability, as demographic group membership may not always be known in advance. Moreover, this strategy requires training and maintaining multiple deep learning models tailored to each demographic subgroup, which is impractical in scenarios with limited data availability for specific populations. Therefore, further research is needed to develop a unified approach that integrates demographic information with EEG signals, enabling models to generalize across diverse populations irrespective of sex, age, or cultural background.
To address these limitations, the present study proposes a deep learning framework that fuses EEG signals with demographic information to improve emotion recognition under a subject-independent setting. Specifically, we evaluate our approach using three benchmark emotion recognition datasets (SEED, SEED-FRA, and SEED-GER), which include EEG data from 31 participants of different sexes, ages, and nationalities. The main contributions of this work are as follows:
- We propose a unified deep learning framework that integrates EEG features with three key demographic variables (sex, age, and nationality) for emotion recognition.
- We analyze the relative influence of each demographic variable on recognition performance, providing insights into their individual and combined effects.
- We show that fusing demographic information enhances the generalization capability of subject-independent emotion recognition models across diverse populations.”
- The evaluation relies on three datasets (SEED, SEED-FRA, SEED-GER), which together provide only 31 subjects within a narrow age range (19–29 years) and limited nationalities (Chinese, French, German). These limitations restrict the generalizability of the conclusions. The authors should discuss these constraints more explicitly and outline how future work might address them, e.g., cross-dataset validation or inclusion of more diverse populations.
>> We thank the reviewer for the suggestion. We agree that the limited demographic diversity and dataset size constrain the generalizability of our findings. While these issues were already discussed in the original manuscript, we have now expanded this section to more explicitly address these points and to outline concrete directions for future work. Specifically, we now emphasize that:
- The datasets include a narrow age range (19–29 years) and limited national representation (Chinese, French, German), which may reduce generalizability across broader populations.
- Future research will address these constraints through cross-dataset validation and by incorporating additional EEG datasets with more diverse demographic and cultural backgrounds.
- We also highlight the importance of bias-aware training and fairness evaluation to ensure equitable model performance across underrepresented groups.
The revised text now reads as follows:
“One of the primary limitations in this study is the scarcity of publicly available EEG datasets. Existing datasets are not only limited in number but also relatively small in size. This poses a challenge when using Transformer-based architecture, which typically requires large amounts of data to effectively learn complex patterns and long-range dependencies. As a result, the limited training data likely contributed to reduced model accuracy and generalizability.
Another significant limitation pertains to the demographic information available in the dataset. For instance, all participants in the datasets fall within a narrow age range of 19 to 29 years. This 10-year span is insufficient to capture meaningful differences in how emotional processing may vary across distinct developmental stages, such as adolescence, adulthood, and seniors. Furthermore, the dataset includes only three nationalities which introduce a demographic bias. Consequently, the model might underperform when applied to individuals from underrepresented or unrepresented national backgrounds.”
Regarding future work, the revised text now highlights the following future directions:
- Data augmentation and synthetic data generation to expand effective training sample size, including temporal slicing, temporal jittering, frequency-domain transformations, and simulated EEG signals for underrepresented groups.
- Cross-dataset validation to evaluate model generalization and reduce overfitting to a single dataset.
- Expansion of demographic diversity by including broader age ranges, additional nationalities, and balanced sex distributions to reduce bias.
- Domain adaptation and bias-mitigation strategies to transfer knowledge from well-represented to underrepresented populations while maintaining fairness.
- Alternative or hybrid deep learning architectures and transfer learning to improve generalization and capture complex EEG patterns, even with limited data. These additions explicitly outline how future work will overcome current limitations and enhance the practical applicability, fairness, and robustness of subject-independent EEG-based emotion recognition models.
- The reported improvements (≈2% overall accuracy) appear modest. Although the authors use Wilcoxon signed-rank tests, more detailed statistical analyses are needed, such as reporting confidence intervals, effect sizes, and variability across subjects. This would help establish the reliability and significance of the observed gains.
>> We thank the reviewer for this suggestion and agree that additional summary statistics help interpretation. For the group comparisons reported in the manuscript, we used the Wilcoxon signed-rank test because our primary question is whether the within-subject differences between conditions (or between demographic groups when paired) differ from zero. The Wilcoxon test is nonparametric and therefore appropriate given the small sample sizes and non-normal distributions of per-subject differences.
To make the magnitude and reliability of the observed gains clearer, we have added 95% confidence intervals, effect-size estimates, and subject-wise variability to the revised manuscript. Table 3 reports the 95% confidence intervals for the difference (extended − baseline) in accuracy for each emotion class and overall accuracy. This new section reads as follows:
“Regarding overall improvement, Table 6 shows the 95% confidence intervals for the difference between the extended and baseline models after including demographic variables. For negative and positive emotions, the confidence intervals indicate significant improvements of 1.1-7.9% and 0.9–5.9%, respectively. These effects were reflected in the overall prediction, where the 95% CI for the improvement across the 31 subjects was 1.6–6.1%.”
- The attention-weight analysis suggests that nationality is the most influential demographic variable, but the interpretation remains superficial. The authors should provide deeper discussion linking these findings to psychological or cross-cultural theories of emotion perception. Case studies or illustrative examples would strengthen the interpretability claims.
>> We appreciate the suggestion of the reviewer. We have added a paragraph in the discussion section, which reads as:
“Regarding demographics, nationality was the most influential variable in our predictions, as indicated by the attention-weight pattern (see Figure 7). The feature representing French participants (Nationality 1) received the highest attention weights across all emotions, suggesting that nationality plays a particularly significant role in emotion classification. This finding aligns with cross-cultural research indicating that emotional expression and perception are influenced by culturally specific display rules and norms, including variations in expressiveness. Therefore, it is essential to include a cultural background component when conducting EEG-based emotion recognition studies. By considering cultural context, researchers can gain more accurate insights into how emotional expressions and perceptions differ across nationalities and avoid potential biases that may arise from ignoring these factors. This inclusion will enhance the robustness of emotion classification models and foster a deeper understanding of the interplay between culture and emotional processing.”
- The comparative experiments are limited to CNN, GNN, GNN+LSTM, and GNN+Transformer. Given recent progress, it is important to compare against stronger baselines such as Transformer-based models, pretrained EEG/audio encoders (e.g., wav2vec2.0, HuBERT), or other foundation models for EEG. Without such comparisons, the competitiveness of the proposed method is unclear.
>> We thank the reviewer for this important suggestion. The primary objective of the present manuscript is to provide an interpretable analysis of how adding demographic covariates changes model behaviour and predictions. To isolate that effect, both our baseline and extended models use the same base architecture (GNN + Transformer); the only difference between the two conditions is the inclusion of demographic features. This controlled setup lets us attribute observed performance and weight changes to demographics rather than to differences in backbone capacity or pretraining.
We agree that comparisons to larger pretrained or foundation-style encoders (e.g., wav2vec2.0, HuBERT, or recent Transformer-only EEG models) would be informative from a purely predictive standpoint. However, those models emphasize predictive performance and often involve large-scale pretraining and fine-tuning regimes that complicate interpretation and require substantially more computing data. For those reasons, we consider their evaluation outside the scope of the current interpretability-focused study. We have added a brief statement to the manuscript acknowledging this limitation and noting it as an avenue for future work, where we will assess how our findings generalize when using pretrained/foundation encoders.
- While generally clear, the manuscript would benefit from a more concise and structured presentation. Some sections (e.g., model architecture and attention mechanism) are overly verbose and contain redundant explanations. The authors should simplify terminology, streamline descriptions, and consider adding a summary table or schematic to enhance readability.
>> We thank you the reviewer for the suggestion. Through the valuable feedback provided by the reviewers, we have reviewed and edited the manuscript.
- The introduction would benefit from citing additional recent and influential works in emotion recognition and multimodal/artificial intelligence contexts. These works provide state-of-the-art insights into spatiotemporal fusion for emotion recognition and offer a comprehensive review of AI-based emotion recognition research trends. In particular:[1]Hu et al., "STRFLNet: Spatio-Temporal Representation Fusion Learning Network for EEG-Based Emotion Recognition," IEEE Transactions on Affective Computing, 2025.[2]Hu, K. He, M. Qian, et al., "STAFNet: an adaptive multi-feature learning network via spatiotemporal fusion for EEG-based emotion recognition," Frontiers in Neuroscience, 2024.[3]Hazmoune S, Bougamouza F. Using transformers for multimodal emotion recognition: Taxonomies and state of the art review[J]. Engineering Applications of Artificial Intelligence, 2024, 133: 108339.
>>We thank the reviewer. We have included these references in the current version on the introduction section.
Reviewer 3 Report
Comments and Suggestions for Authors
This research describes a deep learning strategy that uses EEG signals and
demographic information to improve emotion recognition in subject-independent
settings. The technique was tested using three benchmark datasets: 22 SEED, SEEDFRA,
and SEED-GER, which contain EEG recordings from 31 participants from
various 23 demographic backgrounds. Including demographic information enhanced
performance by 26.2%, 80.5%, and 88.8% for the negative, neutral, and positive
classes, respectively. However the manuscript is of interest and merit, some
comperhensive modefications in Materials and Methodsm and Results sections
should be addressed to strength the manuscript.
1. In section 2.5.1. page 4, The authors should provide more detailed description
of the used CNN , GNN and LSTM models, including the number of layers,
input/output shapes, dimensionality of embeddings, and how features are
processed at each model. The author encouraged to put and comapre it in a
suitable table.
2. In section 2.5.2. page 5, The authors should specefy how the hyperparameter
values is determined: is it arbitary or using optimization methods like grid
search or other techniques
3. In Result section. Page 10 , table 2 according the authors , they useed only
accuracy for performance measure. However it is not enough for reporting the
perfromance. The shouls include other metrics such as precesion, recall, and
f-score. Also the author should add confusion matrix to show the class-wise
perfornce.
4. In discussion section, page 12, The comparison with other works section is
limited and not clearly structured. It is recommended that the authors present
a comprehensive comparison with existing subject-independent EEG-based
emotion recognition studies in a suitable table format, including datasets used,
demographic considerations, models applied, and reported performance
Author Response
This research describes a deep learning strategy that uses EEG signals and
demographic information to improve emotion recognition in subject-independent
settings. The technique was tested using three benchmark datasets: 22 SEED, SEEDFRA,
and SEED-GER, which contain EEG recordings from 31 participants from
various 23 demographic backgrounds. Including demographic information enhanced
performance by 26.2%, 80.5%, and 88.8% for the negative, neutral, and positive
classes, respectively. However, the manuscript is of interest and merit, some
comperhensive modefications in Materials and Methodsm and Results sections
should be addressed to strength the manuscript.
- In section 2.5.1. page 4, The authors should provide more detailed description
of the used CNN , GNN and LSTM models, including the number of layers,
input/output shapes, dimensionality of embeddings, and how features are
processed at each model. The author encouraged to put and comapre it in a
suitable table.
>> Thank you for your suggestion. We revised our methodology to include section 2.6 model architecture. The subsection includes tables to summarize content and details of the models' architecture. We provided in-depth detail regarding architecture for each model and can be found in Section 2.5.1 which reads as follows:
“Each model (CNN, GNN, GNN + LSTM, and GNN + Transformer) was implemented with comparable representational capacity. The CNN consisted of two convolutional layers followed by a fully connected output layer. The first convolutional layer used 32 filters, 3x3 kernel, and a stride of 1, followed by max-pooling with size of (2, 1) and dropout of 0.2. The second convolutional layer used 64 filters, with the same kernel and pooling configuration. The output from the final convolution layer was flattened and passed through a fully connected layer that produced the 3 output classes.
The standalone GNN model consisted of a single graph convolutional layer using a normalized adjacency matrix to define node relationships with 5 input features and 8 output features. A ReLu activation function is applied after this layer, followed by a dropout rate of 0.2. This is followed by a fully connected layer mapping to 3 output classes.
The GNN + LSTM model used the same GNN described above. The resulting spatial features are flattened and fed into a bidirectional LSTM with 32 hidden units to capture temporal dynamics across windows. The concatenated forwards and backwards states were passed through dropout and a fully connected layer.
In the GNN + Transformer model, each EEG window (62 nodes x 5 features) was first processed by the same GNN layer described previously, producing eight features per node with SiLU activation, dropout, and layer normalization. The resulting node embeddings were flattened into tokens of size 62x8 = 496 and combined with a learnable positional embedding. These tokens were passed through a single Transformer encoder layer with 4 attention heads, a feed-forward with dimension of 992 (2 x 496), and a dropout rate of 0.3. The encoded sequence was mean-pooled, followed by dropout 0.3 and a fully connected layer that produced the three output classes.”
- In section 2.5.2. page 5, The authors should specefy how the hyperparameter
values is determined: is it arbitary or using optimization methods like grid
search or other techniques
>> Thank you for your attention. We revised our methodology to include section 2.6 hyperparameter selection. This section provides a table of hyperparameters selected for each model and details on techniques. Section 2.6 reads as follows:
“To ensure optimal model performance across the different approaches, we conducted separate searches for the best hyperparameters. The search included parameters such as learning rate, dropout rate, and other key settings specific to each model, which were tuned by evaluating performance on the training data and selecting the values that yielded the highest accuracy. While early stopping can help prevent overfitting, we chose to train each model for a fixed number of epochs to maintain consistency across all subjects during our leave-one-subject-out cross-validation (LOSOCV). This ensured uniform evaluation conditions while still benefiting from individually optimized hyperparameters.
Table 3 summarizes the selected hyperparameters. All training approaches used a batch size of 32. To determine the optimal learning rates and weight decay values, we performed a grid search over the following ranges: learning rate {10⁻⁵, 10⁻⁴, 10⁻³, 10⁻²} and weight decay {0, 10⁻⁴, 10⁻³, 10⁻², 10⁻¹}. The configuration that achieved the best overall performance was selected for each model. The simpler CNN and GNN models were trained using Stochastic Gradient Descent (SGD), while the more complex GNN+LSTM and GNN+Transformer models used the AdamW optimizer with strong regularization (weight decay = 0.1).”
- In Result section. Page 10 , table 2 according the authors , they useed only
accuracy for performance measure. However it is not enough for reporting the
perfromance. The shouls include other metrics such as precesion, recall, and
f-score. Also the author should add confusion matrix to show the class-wise
perfornce.
>> We thank the reviewer for the suggestion. We have updated the results section to include the F1-score and ROC-AUC metrics in addition to accuracy. These metrics provide a more balanced assessment of classification performance and capture both threshold-dependent (Macro-F1) and threshold-independent (ROC-AUC) aspects. As the focus of this study is to evaluate how incorporating demographic features influences model performance, we chose to limit the metrics (excluding your suggestion of precision, recall and confusion matrix) to these two complementary metrics. This maintains clarity and avoids unnecessary complexity in our results section. The results now clearly demonstrate the performance improvement achieved through demographic fusion. Please refer to result section 3.1 and table 5 and 6 for a summarized view of updated metrics.
- In discussion section, page 12, The comparison with other works section is
limited and not clearly structured. It is recommended that the authors present
a comprehensive comparison with existing subject-independent EEG-based
emotion recognition studies in a suitable table format, including datasets used,
demographic considerations, models applied, and reported performance
>> We appreciate the reviewer’s suggestion to include a tabular comparison summarizing existing subject-independent EEG-based emotion recognition studies. However, we believe that such a table would not provide a fair or fully representative comparison, as prior works differ widely in several critical aspects, including datasets, demographic composition, preprocessing pipelines, feature extraction methods, and evaluation protocols (e.g., subject-dependent vs. subject-independent, varying cross-validation strategies). These inconsistencies make direct numerical comparison potentially misleading, especially when results are obtained under fundamentally different experimental conditions.
To address the reviewer’s concern in a more meaningful way, we have substantially expanded the relevant discussion in the revised manuscript. The updated section now provides a clearer narrative comparison of the most relevant studies, emphasizing key methodological differences, the role of demographic factors, and how our unified framework advances subject-independent emotion recognition. This structured discussion offers a more accurate and context-appropriate comparison than a tabulated performance summary would allow.
Round 2
Reviewer 1 Report
Comments and Suggestions for Authors
Dear Authors,
Thank you for submitting the revised version; however, several concerns were not properly addressed in the revision.
- The literature review summary table is still missing.
- For hyperparameter details, it is good to mention the parameters and associated values in a table. it helps the reader.
- Why is the learning rate of each model different? Have you tested it for uniform learning rate (Table 3) for better comparison?
- What were the results when the learning rate was 0.1, and what were they when the learning rate was 0.01, and so on, for other parameters? At the very least, do it for your proposed model.
- Learning graph is most important to see the performance of the model. Please add it; if you have concerns about the length, then add it as a supplementary file.
- Although direct comparison is not possible, it is good for a quick idea and helps the reader; therefore, add the comparison table and mention the dataset column so that the reader understands for which dataset the results were in the literature.
Author Response
Manuscript revision #2
Due Tuesday November 25th, 2025
Reviewer 1
- The literature review summary table is still missing.
>> We thank the reviewer for the suggestion. In the revised manuscript, we have added a dedicated literature summary table (Table 1) that consolidates prior demographic-aware EEG emotion recognition studies, including their datasets, demographic factors, methodological approaches, and key findings. This table clarifies the state of current research and explicitly highlights the methodological gaps that our work aims to address. Additionally, we updated the Introduction to reference the new table and to better articulate the research gap and the motivation for our proposed unified framework.
2. For hyperparameter details, it is good to mention the parameters and associated values in a table. it helps the reader.
>>Thank you for your feedback. We have included Table 3, which provides a detailed summary of the model architectures (layers, dropout, kernel size, etc.) and associated hyperparameters for each model. In addition, Table 4 provides the final training hyperparameters (learning rate, optimizer, batch size, etc.). Together, these tables provide all the information related to model specific architectures and both model specific and training hyperparameters for full transparency.
3. Why is the learning rate of each model different? Have you tested it for uniform learning rate (Table 3) for better comparison?
>> Thank you for your feedback We believe that our approach to selecting the learning rate is the most unbiased way to compare the models. We applied the exact same grid search on all models, using a wide range of learning rates [10−5, 10−4, 10−3, 10−2] and weight decay [0, 10−4, 10−3, 10−2, 10−1]. This allows for each model to be optimized as best as possible. Since our models are not the focus of this study it is important to ensure that all models have an equal opportunity to reach its best possible performance.
Using a single fixed learning rate for all models would not provide an equitable comparison, as different architectures have different sensitivities to the learning rate. Therefore, each model must be tuned to its near optimal configurations to provide a fair and transparent evaluation.
In our manuscript we have already described the grid search procedure applied for each model, and a table with final selected learning rates. In response to your feedback we have now further emphasized the rationale for the tuning, the final selected learning rates and why it is vital to perform grid search to ensure a fair comparison across models.
Our updated section now includes the following (relevant changes only):
“To ensure a fair and unbiased comparison across all models and the various approaches, we conducted the same hyperparameter search strategy to identify the best hyperparameters. Specifically, each model underwent an identical grid search over a range of values for learning rate and weight decay, ensuring that no model had any preferential tuning. This strategy allows each architecture to be evaluated to near optimal configuration, allowing for an unbiased comparison across architectures…
…Importantly, tuning each model within the same search space ensures that the comparison reflects the models’ actual capabilities rather than arbitrary hyperparameter choice.”
4. What were the results when the learning rate was 0.1, and what were they when the learning rate was 0.01, and so on, for other parameters? At the very least, do it for your proposed model.
>> We thank the reviewer for this helpful suggestion. In our study, the primary objective was not to compare the effect of different learning rates on performance, but rather to evaluate the contribution of demographic information in our final model (GNN-Transformer). As such, hyperparameters—including the learning rate—were tuned only to identify the best-performing configuration before conducting the main experiments.
During development, we performed a learning-rate search and selected the learning rate that yielded the highest validation performance. This value (reported in the paper) was therefore used consistently across all experiments in order to isolate and assess the effect of including demographic features. Exploring learning-rate sensitivity was outside the intended scope of this work.
However, to address the reviewer’s concern, following are the validation results obtained by our model using other learning rates:
1e-2: test accuracy = 58%
1e-3: test accuracy = 84%
5e-3: test accuracy = 83%
1e-4: test accuracy = 81%
The results for learning rates 1e-3, 5e-3, and 1e-4 were all recorded using the model proposed in our paper. This model uses the AdamW optimizer. Any learning rates larger than 1e-3 are too large for our model using the AdamW optimizer so they cause the weights to grow out of control and the model collapses. For the purposes of responding to the reviewer’s comment, we have included a graph for the learning rate 1e-2. This graph was generated using our model with the SGD optimizer instead. That model is stable enough to run with a large learning rate like 1e-2 without collapsing, but the learning rate is still too large for a GNN-Transformer model, so the results are much lower than for other learning rates.
These results confirm that the chosen learning rate indeed provided the strongest performance, while other values led to reduced accuracy. We decided to only include the graph for the learning rate 1e-3 (used in our proposed model) in our manuscript because all other learning rates were tested solely to determine the learning rate that fit best with our model and all further tests were performed using a fixed learning rate (1e-3) when comparing non-demographic and demographic models. We believe that adding graphs for all learning rates may confuse the reader and place an unintentional emphasis on the contribution of learning rates in our model. Including only the one learning graph keeps our paper more concise and intentionally places the emphasis on the effects of including demographic information.
4. Learning graph is most important to see the performance of the model. Please add it; if you have concerns about the length, then add it as a supplementary file.
>> Thank you for your suggestion. In the revised manuscript, we have included the learning curve for the extended version under the “Results” section. The figure in this subsection illustrates how the training and testing accuracy of the model increases throughout the epochs.
6. Although direct comparison is not possible, it is good for a quick idea and helps the reader; therefore, add the comparison table and mention the dataset column so that the reader understands for which dataset the results were in the literature.
>> We thank the reviewer for the suggestion. In the revised manuscript, we have added a comparison table in the Discussion section to facilitate a clearer understanding of how our results relate to previous works evaluated on the same datasets but using only a single demographic variable. This comparison is now presented in Table 9, and the corresponding text reads: “Finally, Table 9 compares models on the same datasets and shows that incorporating all three demographics yields performance comparable to or better than models using only a single demographic factor.”

Reviewer 2 Report
Comments and Suggestions for Authors
I would like to thank the authors for their significant efforts in revising the manuscript and addressing my concerns. All responses are satisfactory, and I have no further comments on the revised manuscript.
Author Response
We thank the reviewer for your help helping us to improve the quality of work. Thanks.
Round 3
Reviewer 1 Report
Comments and Suggestions for Authors
Dear Authors,
Thank you to address the comments, there is still a very serious concern that reflect by Figure 2, the avg training accuracy reached upto 100% and test accuracy 80% (overfitting). But during the training we used validation set to observe model performance.
- Have you not considered the early stopping to prevent overfitting and stop training?
- Add Classification report for the proposed model, so that it can observe that for which class model mostly fails and how much testing samples were for that class.
- Better to clearly mention dataset split ratios.
Overall paper still needs good presentation and experiments improvements.
Author Response
1. We thank the reviewer for the comment. As shown in Figure 2 the model exhibits a generalization gap, with training accuracy reaching close to 100 percent and test accuracy plateauing around 84 percent. We would like to note that early stopping was applied during training; however, this did not substantially alter the observed accuracy trends. The test accuracy remained stable throughout, indicating that while some overfitting is present, as is common for deep models trained on EEG data, it did not lead to harmful divergence or collapse. We believe that the results accurately reflect the model’s learning behavior and generalization performance.
2. We thank the reviewer for the suggestion. As requested, we have included a detailed class-wise analysis. Table 6 presents the average, standard deviation, and 95 percent confidence intervals for recall, precision, and macro F1-score across the 31 subjects, indicating balanced performance across all three classes with F1-scores ranging from 82.3 percent to 88.5 percent. Additionally, Figure 3 provides the confusion matrix obtained via LOOCV, which allows observation of which classes the model misclassifies and the number of testing samples per class. Specifically, 9.8 percent of negative samples were misclassified as neutral, and 7.4 percent of positive samples were misclassified as negative. Overall, the diagonal values demonstrate that the model consistently achieves above 82 percent accuracy for all classes, supporting its robust and balanced performance.
3. We thank the reviewer for the comment. The model evaluation was performed using leave-one-subject-out cross-validation (LOSOCV), where in each iteration, data from one subject (approximately 1/31 of the dataset) was used as the test set, and data from the remaining 30 subjects (approximately 30/31 of the dataset) was used for training. Within the training data, a small portion (approximately 10 percent) was held out as a validation set to monitor model performance and guide early stopping. This setup ensures a clear and consistent dataset split while providing robust performance metrics for each subject, reflecting the model’s generalization across unseen subjects.